# Development of New Open-Set Speech Material for Use in Clinical Audiology with Speakers of British English

**Mahmoud Keshavarzi** [1,2,3], **Marina Salorio-Corbetto** [2,4,5], **Tobias Reichenbach** [1,6], **Josephine Marriage** [4] **and Brian C. J. Moore** [2,*]

1. Department of Bioengineering and Centre for Neurotechnology, Imperial College London, London SW7 2AZ, UK; mk919@cam.ac.uk (M.K.); tobias.j.reichenbach@fau.de (T.R.)
2. Cambridge Hearing Group, Department of Psychology, University of Cambridge, Cambridge CB2 3EB, UK; ms878@cam.ac.uk
3. Centre for Neuroscience in Education, Department of Psychology, University of Cambridge, Cambridge CB2 3EB, UK
4. Chear Ltd., Royston SG8 6QS, Herts, UK; josephine@chears.co.uk
5. Cambridge Hearing Group, Sound Laboratory, Department of Clinical Neurosciences, University of Cambridge, Cambridge CB2 0QQ, UK
6. Department Artificial Intelligence in Biomedical Engineering, Friedrich-Alexander-University Erlangen-Nuremberg, 91052 Erlangen, Germany
* Correspondence: bcjm@cam.ac.uk

**Abstract:** Background: The Chear open-set performance test (COPT), which uses a carrier phrase followed by a monosyllabic test word, is intended for clinical assessment of speech recognition, evaluation of hearing-device performance, and the fine-tuning of hearing devices for speakers of British English. This paper assesses practice effects, test–retest reliability, and the variability across lists of the COPT. Method: In experiment 1, 16 normal-hearing participants were tested using an initial version of the COPT, at three speech-to-noise ratios (SNRs). Experiment 2 used revised COPT lists, with items swapped between lists to reduce differences in difficulty across lists. In experiment 3, test–retest repeatability was assessed for stimuli presented in quiet, using 15 participants with sensorineural hearing loss. Results: After administration of a single practice list, no practice effects were evident. The critical difference between scores for two lists was about 2 words (out of 15) or 5 phonemes (out of 50). The mean estimated SNR required for 74% words correct was −0.56 dB, with a standard deviation across lists of 0.16 dB. For the participants with hearing loss tested in quiet, the critical difference between scores for two lists was about 3 words (out of 15) or 6 phonemes (out of 50).

**Keywords:** speech test; open-set speech material; word analysis; phoneme analysis

## 1. Introduction

Although the primary purpose of hearing aids (HAs) and cochlear implants (CIs) is to improve the understanding of speech in quiet and in background noise [1–4], speech testing is not consistently used in the fitting and verification of HAs [5] and is not always used in the fitting of CIs. Speech tests may be useful for verifying that a fitting is adequate, quantifying the benefit of HAs and CIs [6], assessing the effectiveness of novel signal-processing strategies, demonstrating any benefit for speech perception to the patient, and also for assessing candidacy for CIs [7,8]. The National Institute for Health Care Excellence in the UK currently recommends speech testing as part of the assessment of adults for candidacy for CIs [9].

Two reasons that speech tests are not always used in the fitting and evaluation of HAs and CIs, at least in the UK, are the lack of time to administer sufficient test items to obtain an accurate outcome [10,11] and the limited range and type of available tests. With closed-set tests, the listener selects from a limited number of response alternatives. Examples are

the consonant confusion test (CCT) and the Chear Auditory Perception Test (CAPT), both of which were designed for use with young children [12]; the speech pattern contrast test [13]; and various "matrix" tests that have been developed for use with adults in many different languages [14–16]. A disadvantage of matrix tests is that the predictable structure of the materials often leads to very low speech reception thresholds in the presence of background noise or competing talkers (SRTs; the speech-to-background ratio required to achieve a given percentage correct) [17], lower than would typically be encountered in everyday life [18]. Many of the signal-processing algorithms in HAs and CIs do not work effectively when the speech-to-background ratio is very low [3], so SRTs measured with matrix tests may not give a realistic estimate of the performance of those algorithms under everyday conditions.

Open-set speech tests overcome some of these problems. With open-set tests, the listener simply repeats what has been heard, rather than selecting from a limited number of response alternatives. The use of open-set materials increases the face validity of the tests and leads to higher and more realistic SRTs. To maintain the validity of open-set speech tests, the materials should not be familiar to the listener and no sentence should be repeated (if a word or a list has been heard previously during testing, the listener is more likely to make correct responses than when a word/list has not been heard). This means that many lists of similar difficulty must be available to allow different conditions to be compared and to allow testing at several time points, for example in longitudinal studies.

Examples of open-set materials are the sentence test, described by Plomp and Mimpen [19] for the Dutch language, and the Hearing-in-Noise Test (HINT) [20] and the QuickSIN test [21], both of which use talkers with American accents. This is important since accents/dialects can affect the intelligibility of speech, especially at low speech-to-noise ratios [22]. Among the open-set materials that are used in the UK with adults are the BKB sentences (21 lists) [23] and the AB word lists (15 lists) [24]. Both sets of materials have limitations when used to evaluate signal processing in HAs and CIs, as is described next. Many of the signal-processing algorithms in HAs and CIs operate over relatively long time scales. As an example, many HAs and CIs incorporate slow-acting automatic gain-control systems, in which the gain changes slowly over time [25–27]. The release time constants for such systems can reach or exceed 1000 ms. Hence, when an item from a speech test is presented, the gain may change during the presentation of the item, and for a short item, such as an isolated word, the gain that is applied may not be representative of the gain that would be applied to running speech. Similarly, some noise-reduction systems in HAs and CIs sample the signal for some time to analyse the patterns of amplitude modulation in different frequency regions and then reduce the gain in frequency regions where the amount of amplitude modulation is low and noise rather than speech is probably dominant [28,29]. The time constants of such systems vary widely across manufacturers [30]. This means that the amount of noise reduction may vary during the presentation of speech-test material, and for a short test item, the noise reduction that is applied may not be representative of what would occur during running speech.

Some speech tests involve the use of a carrier phrase such as "Please say the word . . ." prior to the test word. Examples are the NU6 test [31] and the Words-In-Noise (WIN) test [32]. The use of a carrier phrase allows the signal-processing parameters of an HA or CI, such as gain and noise reduction, to settle at least partially before the target word is presented. The use of a set carrier phrase also allows the listener to become familiar with the characteristics of the talker, as would usually be the case in everyday life. However, for both the NU6 and the WIN test, the talker has an American accent. Also, the NU6 test includes only four lists, while the WIN test includes only 70 words (10 words at each of seven signal-to-noise ratios, SNRs). We are not aware of a test with a carrier phrase for which the talker has a British accent and for which there are enough lists to allow performance to be assessed over time (e.g., before and after being fitted with a hearing device) without repeating a list. This paper describes a new open-set speech test that includes a carrier phrase and uses a talker with a British accent. The test is called the

Chear open-set performance test (COPT). "Chear" is the name of the clinical practice where the test was developed. In each trial, the COPT uses a carrier phrase followed by a monosyllabic test word. Time constraints in the clinic limit the number of items that can be used for each list, but the number of words needs to be sufficient to give reliable and reproducible estimates of intelligibility. For the COPT, the number of words per list was chosen as 15, as a compromise between these requirements. The typical test time for each list is about 3 min. Both phoneme and word scoring can be used with the COPT. Open-set phoneme scoring can be more effective in identifying changes in hearing than open-set word scoring or closed-set word scoring [33]. The COPT materials can be presented in background noise or in quiet. This paper focuses on the former, but some preliminary data for the latter are also presented. The main purposes of this paper are to describe the COPT and to assess practice effects, the consistency of scores across lists, the repeatability across lists, and the slope of the psychometric function, i.e., the function relating performance to SNR, using normal-hearing adults. People with hearing loss would usually try to avoid trying to converse in situations where they only get 50% correct. Therefore, we decided to assess performance using SNRs where performance was reasonably high, to reflect realistic listening conditions. This had two added advantages: (1) the SNRs used were in the range where signal-processing algorithms in HAs and CIs, such as noise reduction, work most effectively; (2) high but not perfect scores help to maintain the concentration and motivation of the participants, preventing them from "giving up" [34]. Since the COPT may also be used to assess speech perception in quiet, preliminary data were also obtained on test–retest repeatability for participants with hearing loss tested in quiet.

## 2. Method for Experiments 1 and 2

Experiment 1 evaluated an initial version of the COPT. Experiment 2 evaluated a revised version of the COPT for which items were swapped across lists to make performance more similar across lists. These experiments assessed performance for the COPT lists presented in background noise.

### 2.1. Participants

A total of 16 native speakers of British English (10 women and 6 men, average age = 24 years, standard deviation = 5.4 years, age range 19–36 years) were tested in experiment 1. A different group of 15 native speakers of British English (8 women and 7 men, average age = 24.1 years, standard deviation, SD = 4.7 years) were tested in experiment 2. Audiometric thresholds were measured for audiometric frequencies from 0.125 to 8 kHz, using an Amplivox audiometer (Amplivox, Birmingham, UK). All participants in experiments 1 and 2 had normal hearing, with audiometric thresholds ≤20 dB HL for all measured frequencies, and only the better-hearing ear of each participant was tested (based on the average threshold across 0.5 to 4 kHz). The test session lasted for about 1.5 h for each participant. Participants were paid for taking part and their travel expenses were reimbursed. This study was approved by the Imperial College Research Ethics Committee.

### 2.2. Speech Material and Noise

Two sets of speech materials were used. The first set was used to select appropriate SNRs for each participant in the main experiments with the COPT materials, to give SNRs leading to high but not perfect performance. The first set was chosen from an English multi-talker corpus named GRID [35]. Sentences were of the form "Put *__red__* at *__G 9__* now", where the target words are underlined and in bold italic font. One hundred and twenty sentences from eight different talkers (four men and four women) were used.

The second set of test materials were the lists from the COPT. There were 16 lists, each of which had 15 target words, giving 16 × 15 = 240 different words in total. Words were recorded with the preceding carrier phrase "Say the word", giving sentences such as "Say the word gasp". Within each list, the order of the words was fixed. The carrier phrase was spoken before each test word, so natural coarticulation effects were present. The carrier

phrase was also intended to allow any signal processing in hearing devices to settle, at least partially. On average, the test word started 0.78 s after the start of the carrier phrase, with an SD of 0.09 s. All words were monosyllabic and they were chosen to have several similar sounding real-word lexical alternatives [36]. For example, the test word "thick" was potentially confusable with "sick", "chick", and "fit". Since there are 44 phonemes in British English, and, on average, there were 50 phonemes per list, it was not possible to achieve full phonemic balance. However, each list had approximately equal numbers of plosives, fricatives, glides, nasals, and affricates. All lists included some common words and some less common words, such as "hawks" and "smock".

The talker was a female native speaker of British English with a received-pronunciation accent. The talker spoke with normal vocal effort and articulated clearly. The speech level was monitored with a CEL 414 precision sound level meter, set to A weighting and "slow" averaging, placed about 50 cm from the talker and visible to the talker. The talker aimed to keep the meter reading as constant as possible, and peak levels were typically kept within a range of $\pm 2$ dB. The talker was seated in a double-walled sound-attenuating booth designed to have minimal reflections and low background noise. The microphone was a high-quality studio model (Sennheiser MKH 40 P 48 U3, Marlow, UK). The microphone was approximately 30 cm from the mouth of the talker in the horizontal direction and 15 cm below the mouth of the talker and faced the talker's mouth. The output of the microphone was digitised with 24-bit resolution and recorded on the hard drive of a PC. For this study, the recordings were down-sampled from 44.1 to 16 kHz to reduce data storage requirements. This allowed the representation of frequencies up to almost 8 kHz, which is the most important range for speech perception [37] and which covers the frequency range provided by most hearing aids [38] and cochlear implants [39]. Each carrier phrase plus test word was recorded several times (usually five), and the exemplar with an overall level closest to the mean level was chosen for further use. We wanted to preserve the natural level of each test word relative to the carrier phrase and to avoid discontinuities in the transition from carrier phrase to test word. Therefore, no adjustment was made to the relative levels of individual test words to reduce differences in level or difficulty of the individual words.

The background noise used with the GRID materials had the same long-term average spectrum as the 120 sentences. The background noise used with the COPT materials had the same long-term average spectrum as the target words. In each trial, the noise started synchronously with the start of the sentence or carrier phase and ended 0.25 s after the end of the sentence or target word. A new noise sample was used for each trial.

*2.3. Procedure*

The participant was seated in a sound-attenuating room and wore Sennheiser HD215 headphones connected to the sound card of a MacBook Pro. Each experiment had two parts. In the first part, an adaptive procedure was used to determine the SNR leading to 66.6% words correct for each participant, using the GRID material. This is denoted $SRT_{GRID}$. The speech and noise were stored as separate files and the level of each was adjusted using custom software prior to mixing. The speech was presented at 65 dB SPL and the noise level was varied to adjust the SNR. In each trial, a randomly selected sentence from the GRID material was presented. The instructions to the participant, which appeared on the computer screen, were as follows: "On each trial you will hear one sentence in which a colour, a letter, and a number are mentioned. Please determine what colour, what letter and what number you hear in each trial". After the sound had been presented, the following instructions appeared on the computer screen: "Please select the colour, and then put the letter and number in each box separately". The colour options of "Red", "Blue", "Black", "Yellow", "Green", and "White" were shown below the instructions on the screen. There were also two blank boxes below the colour options for the number and letter. Responses were entered on a keyboard.

The initial SNR was randomly chosen between 0 dB and −3 dB. If the participant correctly reported at least two out of three key words (letter, colour, and number) in a sentence correctly, the SNR was decreased by 1 dB for the subsequent trial; otherwise, it was increased by 1 dB. This tracks the SNR leading to 66.6% correct [40]. The adaptive procedure was stopped after seven reversals in the SNR or after 17 sentences, whichever occurred sooner. The adaptive procedure was repeated four times for each participant and the final value of $SRT_{GRID}$ was computed as the average of the SNRs that were employed in the last three runs. The individual and mean values of $SRT_{GRID}$ for experiment 1 are shown in Table 1. Individual differences in $SRT_{GRID}$ were small.

**Table 1.** Results of experiment 1 showing individual $SRT_{GRID}$ values and percentage of words correct for the COPT for the three SNRs separated by 3 dB: low, moderate, and high. The means and SDs of the scores are shown at the bottom.

| Participant | $SRT_{GRID}$, dB | Percent Correct Low SNR | Percent Correct Moderate SNR | Percent Correct High SNR |
|:---:|:---:|:---:|:---:|:---:|
| 1 | −3.3 | 44.4 | 80.0 | 84.4 |
| 2 | −3.0 | 68.9 | 75.6 | 88.9 |
| 3 | −3.0 | 62.2 | 82.2 | 88.9 |
| 4 | −3.3 | 55.6 | 75.6 | 82.2 |
| 5 | −2.6 | 71.1 | 80.0 | 88.9 |
| 6 | −2.8 | 62.2 | 66.7 | 91.1 |
| 7 | −2.5 | 55.6 | 80.0 | 86.7 |
| 8 | −2.5 | 73.3 | 86.7 | 95.6 |
| 9 | −3.0 | 66.7 | 71.1 | 93.3 |
| 10 | −1.5 | 66.7 | 84.4 | 93.3 |
| 11 | −2.9 | 57.8 | 82.2 | 93.3 |
| 12 | −2.6 | 62.2 | 75.6 | 82.2 |
| 13 | −2.4 | 64.4 | 75.6 | 93.3 |
| 14 | −2.5 | 64.4 | 84.4 | 97.8 |
| 15 | −3.1 | 75.6 | 84.4 | 93.3 |
| 16 | −3.2 | 77.8 | 82.2 | 88.9 |
| Mean | −2.8 | 64.3 | 79.2 | 90.1 |
| SD | 0.4 | 8.2 | 5.3 | 4.4 |

In the second part of experiment 1, each participant listened to ten lists chosen from the COPT word lists. The speech was presented at 65 dB SPL and the noise level was varied to adjust the SNR. For each participant, one list was presented at the SNR corresponding to $SRT_{GRID}$, as determined in part 1. This was treated as a practice list and was not scored. Then, nine more lists were presented, three at $SRT_{GRID}$ ($SNR_{LOW}$), three at $SRT_{GRID}$ + 3 dB ($SNR_{MODERATE}$), and three at $SRT_{GRID}$ + 6 dB ($SNR_{HIGH}$). The order was low, moderate, high, low, moderate, high, low, moderate, high. This ordering was chosen to reduce fatigue effects. The starting list was rotated across participants, such that each test list was presented nine times (three at $SNR_{LOW}$, three at $SNR_{MODERATE}$, and three at $SNR_{HIGH}$), based on results from nine different participants. In the second part of experiment 2, the procedure was similar except that, apart from one list that was used for practice, all 15 participants were tested using all fifteen lists, with the list order and SNR rotated across participants.

In each trial, a single sentence was presented. The following instructions appeared on the computer screen: "On each trial you will hear a phrase "**Say the word . . .**". You will then be asked to write the word the speaker says". After the sound had been presented, the following instructions appeared on the computer screen: "Please write the word that you heard". There was a blank box on the screen for the word to be written in, using a keyboard. This procedure was preferred to the experimenter recording the oral responses of the participant to avoid the possibility of the experimenter mis-hearing the participant. No feedback was provided. Scoring was performed by a native speaker of

English and homophones (e.g., responding "scene" when the test word was "seen") were scored as correct.

## 3. Results of Experiment 1

### 3.1. Word-Level Analyses

For the word-level analyses, the response on each trial was assigned a score of 1 if the word (all phonemes) was correctly identified; otherwise, the score was 0. To assess whether there were practice effects, scores were compared for the first list used with a given participant, which was always presented at $SNR_{LOW}$, and for the fourth list used with a given participant, which was also presented at $SNR_{LOW}$. The mean scores across the 16 participants for these two cases were identical at 66.3% correct. Hence, there was no evidence for practice effects (but recall that a practice list presented at $SNR_{LOW}$ was administered prior to testing proper).

Table 1 shows the individual and mean values of $SRT_{GRID}$ (in dB) and the scores for each SNR for the COPT. For further analysis, the percent correct scores for each participant, each SNR, and each list were converted to Rationalised Arcsine Units (RAUs). This has the effect of approximately linearising the psychometric function relating performance to SNR and of making the distribution of scores close to normal [41,42].

To obtain an overall estimate of the slope of the psychometric function relating RAU scores to SNR, the following procedure was adopted.

(1)  For each participant, the RAU scores were averaged for each SNR across the five lists used with that SNR.
(2)  A straight line was fitted to the RAU values versus SNR for each participant.
(3)  The slope of the fitted line was averaged across participants.

The resulting slope was 4.75 RAU/dB. In other words, on average, the score increased by 4.75 RAU for each 1 dB increase in SNR. For $SNR_{MODERATE}$, the mean score was about 79 RAU correct, which corresponds roughly to 79% correct. Because the psychometric function is typically S-shaped when scores are expressed in percent, the slope in percent/dB around 79% correct is shallower than the slope around 50%, which has often been used in previous studies [19].

To obtain an initial estimate of the extent to which overall difficulty varied across lists, we calculated the mean percentage of words correct separately for each list but averaged across the three SNRs. The mean score varied from about 70% to 85% correct across lists, with a mean of 78% and an SD of 4.5%. Some of the variability across lists may have occurred because not all lists were used with all participants, and the values of $SRT_{GRID}$ varied across participants, as shown in Table 1.

The following procedure was adopted to assess the extent to which difficulty varied across lists when compared at similar SNRs.

(1)  For each list (x = 1–16) and each nominal SNR, the percent correct scores were converted to RAU scores. The resulting RAU scores were averaged and the actual SNRs used were also averaged. This gave three data points for each list (one for each nominal SNR), relating RAU score to actual SNR for that list.
(2)  For each list, a linear function was fitted to the three points (the three RAU scores for each of three actual SNRs). This function is characterised by its slope, S(x), and its intercept, I(x).
(3)  The actual SNR required to give 77.85 RAU (corresponding to 80% correct with 15 words per list) was calculated from S(x) and I(x). This SNR is denoted SRT(x); it represents the average SRT for list x. The value of 80% correct is arbitrary, but it was chosen to be close to the mean word score for $SNR_{MODERATE}$. The values of SRT(x) are plotted in Figure 1.
(4)  The SRT(x) values were averaged across all lists (all values of x). The resulting SNR is denoted SRT(Ave); it represents the SNR required to achieve 77.85 RAU averaged across all lists. The value of SRT(Ave) was 0.2 dB.

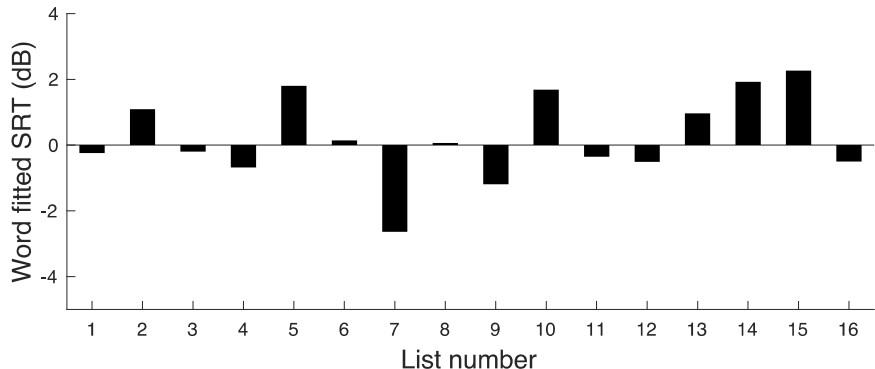

**Figure 1.** Results of experiment 1, showing SRT values corresponding to a word score of 77.85 RAU for each list.

The deviations in SRT(x) from SRT(Ave) give a measure of the variability in difficulty across lists. The deviations ranged from $-2.9$ to 2.0 dB. The SD of the SRT(x) values was 1.3 dB. This variability is somewhat larger than would be desired.

*3.2. Phoneme-Level Analyses*

For the phoneme-level analyses, a score of 1 was assigned for each correctly reported phoneme. For example, if the participant's response to "lake" was "cake", the score was 2 (out of 3). To assess whether there were practice effects, scores were compared for the first list used with a given participant, which was always presented at $SNR_{LOW}$, and for the fourth list used with that same participant, which was also presented at $SNR_{LOW}$. The mean scores across participants for these two cases were almost the same at 84.6% and 85.2% correct. A matched-samples *t*-test showed that the difference was not significant. Hence, there was no evidence for practice effects.

Table 2 shows the individual and mean values of the scores for each SNR. To obtain an overall estimate of the slope of the psychometric function relating RAU scores to SNR, the same procedure as for word scores were adopted, except that the RAU scores were based on 50 phonemes per list.

**Table 2.** As Table 1 but showing the percentage of phonemes correct.

| Participant | Percent Correct Low SNR | Percent Correct Moderate SNR | Percent Correct High SNR |
|:---:|:---:|:---:|:---:|
| 1 | 75.8 | 91.3 | 95.4 |
| 2 | 86.7 | 95.9 | 96.0 |
| 3 | 86.3 | 94.0 | 97.3 |
| 4 | 79.5 | 90.7 | 90.8 |
| 5 | 88.7 | 93.4 | 95.3 |
| 6 | 84.2 | 86.7 | 97.3 |
| 7 | 83.3 | 92.0 | 95.4 |
| 8 | 89.3 | 94.7 | 98.0 |
| 9 | 88.8 | 90.0 | 97.3 |
| 10 | 88.0 | 93.3 | 98.0 |
| 11 | 82.6 | 93.4 | 98.0 |
| 12 | 82.8 | 90.0 | 94.0 |
| 13 | 83.3 | 88.6 | 98.0 |
| 14 | 83.2 | 94.0 | 99.3 |
| 15 | 90.7 | 94.7 | 98.0 |
| 16 | 90.8 | 94.0 | 96.7 |
| Mean | 85.2 | 92.3 | 96.6 |
| SD | 4.0 | 2.5 | 2.0 |

The resulting slope was 3.1 RAU/dB. Converting back to percentage scores, this is equivalent to a slope of about 2% per dB. Again, it should be emphasised that this is the slope in the region of the psychometric function where scores are high, in the region of 92% correct.

To get an initial estimate of the extent to which overall difficulty varied across lists, we calculated the mean percentage of phonemes correct separately for each list but averaged across the three SNRs. The mean score varied from about 87% to 94.5% correct across lists, with a mean of 91% and an SD of 2.2%. The same procedure as used for the word scores was adopted to assess the extent to which difficulty varied across lists when compared at similar SNRs. However, for the phoneme analyses, the values of SRT(x) were calculated as the SNR leading to 95 RAU (corresponding to 92% correct). This is close to the mean percentage score for the moderate SNR. The values of SRT(x) based on phoneme scores are plotted in Figure 2.

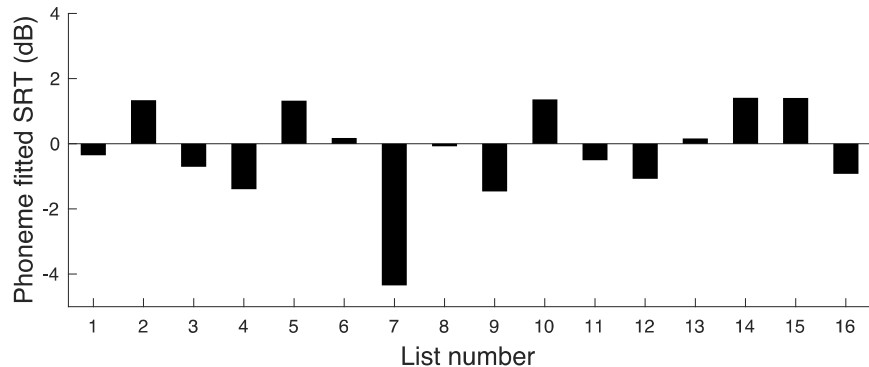

**Figure 2.** Results of experiment 1, showing SRT values corresponding to a phoneme score of 95 RAU for each list.

The value of SRT(Ave) was −0.2 dB. The deviations in SRT(x) from SRT(Ave) give a measure of the variability in difficulty across lists. The deviations ranged from −4.1 to 1.6 dB. The SD of the SRT(x) values was 1.5 dB.

## 4. Experiment 2

### 4.1. Procedure for List Adjustment

Adjustments were made to the COPT lists to make the difficulty more equal across lists. The procedure was as follows:

(1)  The percent correct score was calculated for each word in each list used in experiment 1, based on the scores for the nine participants (three for each SNR) tested with each list. For example, if the word "wrist" was reported correctly by seven out of nine participants, the percent correct score for that word was 77.8.
(2)  List 1 was designated as the practice list.
(3)  The percent correct scores were calculated for each of lists 2–16 by averaging the percent correct scores for the 15 words in each list.
(4)  Words with very low scores (between 0% and 11.1%) were moved from lists 2–16 to the practice list and replaced by words with higher scores from the practice list.
(5)  Words were swapped between lists 2–16 with the following goals:

(a)  To equate the percent correct scores as closely as possible across lists 2–16.
(b)  So that each list in lists 2–16 would include 50 phonemes.
(c)  So that after the swapping, it was still the case that each list had approximately equal numbers of plosives, fricatives, glides, nasals, and affricates.

As a result of this procedure, two lists had no words changed, six lists had one word changed, three lists had two words changed, and four lists had three words changed. The words in the 15 test lists are given in Table 3. The calculated mean percent correct score (based on the results of experiment 1) for the adjusted lists was 80.1% with an SD across

lists of 1.7%, so, in theory at least, goal (a) was achieved. Goals (b) and (c) were achieved for all 15 test lists.

**Table 3.** Test words for the 15 test lists used in experiment 2.

| | | | | | | | List Number | | | | | | | |
|---|---|---|---|---|---|---|---|---|---|---|---|---|---|---|
| **2** | **3** | **4** | **5** | **6** | **7** | **8** | **9** | **10** | **11** | **12** | **13** | **14** | **15** | **16** |
| chops | tent | hide | leaf | doubt | lean | wheel | weep | weed | torch | talk | mouth | part | thorn | lock |
| news | tool | pokes | coke | suit | skip | move | flute | loop | root | stores | sell | track | roof | great |
| belt | leave | leaps | vest | thief | laugh | cope | need | seat | knees | gnat | deep | seed | sheet | kept |
| bite | gap | check | heap | bike | mine | wick | knife | pride | fort | beak | creep | treat | team | fight |
| gasp | thirst | short | chalk | tread | stalk | fake | caught | tall | died | tag | right | ripe | rhyme | chime |
| moth | maze | rest | roam | taps | soup | crouch | tease | loans | phone | claws | flows | mark | boast | taught |
| beg | feed | soap | web | wit | cog | best | desk | shine | weak | nest | step | shell | fed | wash |
| whip | which | whim | bed | make | meet | thick | bring | takes | wrist | wish | thin | clues | skin | seal |
| kiss | place | plate | steak | scarf | tail | dates | gate | case | capes | gaze | cage | perch | rule | smock |
| sharp | tart | dark | scars | mauve | start | stars | shark | guard | flake | tape | game | days | gates | stick |
| pack | wreck | snag | cone | rank | flag | hike | tanned | fact | snap | smack | drag | snack | trap | rag |
| let | smoke | debt | felt | hell | peep | mop | fetch | neck | head | page | turn | gave | pest | peaks |
| knock | soak | north | torn | height | must | mug | stone | led | tide | lied | set | made | male | lake |
| sent | bun | moss | get | knob | shock | rocked | wipe | stop | pots | race | pop | newt | cod | coach |
| crumb | kicks | thorn | truck | speech | beach | teeth | shoes | peach | mean | bloom | tossed | cost | porch | rose |

### 4.2. Values of SRT_GRID

The individual values of $SRT_{GRID}$ for experiment 2 are shown in Table 4. Individual differences in $SRT_{GRID}$ were again small.

**Table 4.** As Table 1 but for word scores for experiment 2.

| Participant | $SRT_{GRID}$, dB | Percent Correct Low SNR | Percent Correct Moderate SNR | Percent Correct High SNR |
|---|---|---|---|---|
| 1 | −3.4 | 62.7 | 77.3 | 82.7 |
| 2 | −3.3 | 64.0 | 76.0 | 82.7 |
| 3 | −3.3 | 64.0 | 74.7 | 90.7 |
| 4 | −3.5 | 65.3 | 80.0 | 93.3 |
| 5 | −3.0 | 65.3 | 76.0 | 89.3 |
| 6 | −3.7 | 66.7 | 77.3 | 88.0 |
| 7 | −2.8 | 62.7 | 72.0 | 86.7 |
| 8 | −3.4 | 61.3 | 73.3 | 86.7 |
| 9 | −3.7 | 56.0 | 70.7 | 86.7 |
| 10 | −3.9 | 60.0 | 77.3 | 90.7 |
| 11 | −3.4 | 64.0 | 74.7 | 88.0 |
| 12 | −2.6 | 60.0 | 72.0 | 86.7 |
| 13 | −3.7 | 62.7 | 73.3 | 86.7 |
| 14 | −2.7 | 68.0 | 77.3 | 88.0 |
| 15 | −2.8 | 68.0 | 76.0 | 93.3 |
| Mean | −3.3 | 63.4 | 75.2 | 88.0 |
| SD | 0.4 | 3.2 | 2.5 | 3.0 |

### 4.3. Word-Level Analyses

Table 4 shows the scores for each SNR for the COPT for the 15 test lists. The means were similar to those for experiment 1, but the SDs were markedly smaller than those for experiment 1, indicating that the adjustments were effective in making scores more similar across lists.

To obtain an overall estimate of the slope of the psychometric function relating RAU score to SNR, the same procedure as for experiment 1 was adopted. The resulting slope was 4.4 RAU/dB, similar to the slope found in experiment 1.

To assess the test–retest reliability associated with application of a single test list, critical differences were calculated. For each SNR, the critical difference was computed as $\sqrt{2} \times 1.96 \times \sqrt{V}$, where V is the mean variance across the individual variances of the RAU scores for each participant [43,44]. The average critical difference across SNRs was 14.9 RAU. The critical differences varied with SNR, being 12.1, 14.4, and 18.0 RAU for the low, moderate, and high SNRs, respectively. This means that if two conditions were compared using a single list for each condition, scores for two conditions would need to differ by two words for the difference to be considered significant, regardless of the SNR. For two conditions compared using two lists for each condition, the critical differences were 10.5, 12.9, and 11.9 RAU for the low, moderate, and high SNRs, respectively, with an average across SNRs of 11.9 RAU. These correspond to a difference in word scores of 3 (out of 30).

To obtain an estimate of the extent to which overall difficulty varied across lists, a similar procedure to that for experiment 1 was adopted. Specifically:

(1) Percent correct scores were converted to RAU scores for each list, participant, and SNR.
(2) The RAU scores were averaged for each list for a given SNR across the five participants used with that SNR and list.
(3) For each list, the mean RAU score was calculated for the middle SNR. This is denoted RAU(Ave). Its value was 72.9 RAU.
(4) A straight line was fitted to the mean RAU values versus SNR for each list.
(5) The fitted line was used to estimate the SNR required for each list to obtain RAU(Ave). The result is the word SRT for each list.

The resulting SRTs are shown in Figure 3. The mean SRT for the word scores was SRT(words) = −0.56 dB and the SD was 0.16 dB. This SD is much smaller than the SD of 1.3 dB for the first version of the COPT, indicating that the adjustments were successful in markedly reducing the variability in difficulty across lists.

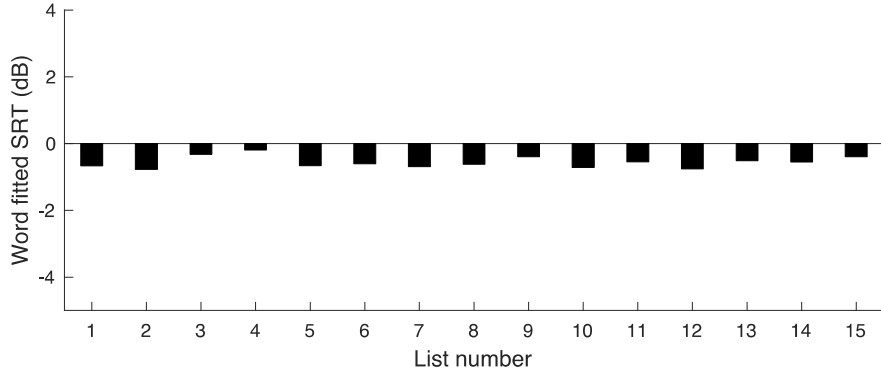

**Figure 3.** Results of experiment 2, showing SRT values corresponding to a word score of 72.9 RAU for lists 2–16.

### 4.4. Phoneme-Level Analyses

The mean phoneme score was calculated separately for each SNR condition based on the 75 words (250 phonemes) used with that condition. Table 5 shows the individual and mean values of the scores for each SNR. The average slope of the linear function relating RAU score to SNR was 3.4 RAU/dB.

To assess the test–retest reliability associated with the application of a single test list, critical differences were calculated following the same method as described for the word-level analysis, based on RAU scores. The average critical difference across SNRs was 11.7 RAU. The critical differences varied with SNR, being 11.6, 9.5, and 14.1 RAU for the low, moderate, and high SNRs, respectively, with a mean across SNRs of 11.7 RAU. These correspond to differences in percent correct of about 9.9, 6.4, and 6.5, respectively, i.e., about 5, 3, and 3 phonemes, respectively. This means that if two conditions were compared using a single list for each condition, scores for two conditions would need to differ by about

5 phonemes for scores close to 82%, and 3 phonemes for scores in the range from 90% to 96%. For two conditions compared using two lists for each condition, the critical differences were 10.9, 10.9, and 9.8 RAU for the low, moderate, and high SNRs, respectively, with a mean across SNRs of 10.5 RAU. These correspond to a difference in phoneme scores of about 7 (out of 100).

**Table 5.** As Table 2 but for phoneme scores for experiment 2.

| Participant | Percent Correct Low SNR | Percent Correct Moderate SNR | Percent Correct High SNR |
|---|---|---|---|
| 1 | 76.8 | 92.0 | 92.4 |
| 2 | 82.8 | 90.4 | 93.6 |
| 3 | 84.8 | 90.4 | 96.8 |
| 4 | 84.8 | 91.6 | 97.6 |
| 5 | 82.0 | 90.4 | 96.4 |
| 6 | 84.4 | 89.6 | 96.4 |
| 7 | 82.0 | 86.8 | 95.2 |
| 8 | 82.4 | 87.2 | 95.2 |
| 9 | 78.8 | 87.2 | 94.4 |
| 10 | 80.0 | 89.6 | 96.4 |
| 11 | 84.8 | 90.8 | 95.6 |
| 12 | 82.4 | 89.6 | 96.0 |
| 13 | 80.0 | 89.2 | 94.4 |
| 14 | 85.2 | 90.4 | 95.2 |
| 15 | 84.4 | 90.8 | 98.0 |
| Mean | 82.4 | 89.7 | 95.6 |
| SD | 2.4 | 1.5 | 1.4 |

The same procedure as used for the word scores was adopted to assess the extent to which difficulty varied across lists. The SRTs based on phoneme scores are plotted in Figure 4. The value of SRT(Ave) was −0.38 dB. The deviations of the SRTs from SRT(Ave) give a measure of the variability in difficulty across lists. The SD of the SRTs was 0.39 dB. The SD was markedly smaller than for the phoneme-based SRTs of experiment 1, indicating that the adjustments made were successful in reducing the variability across lists. However, the SD was larger than for the SRTs for word scores, which was only 0.16 dB.

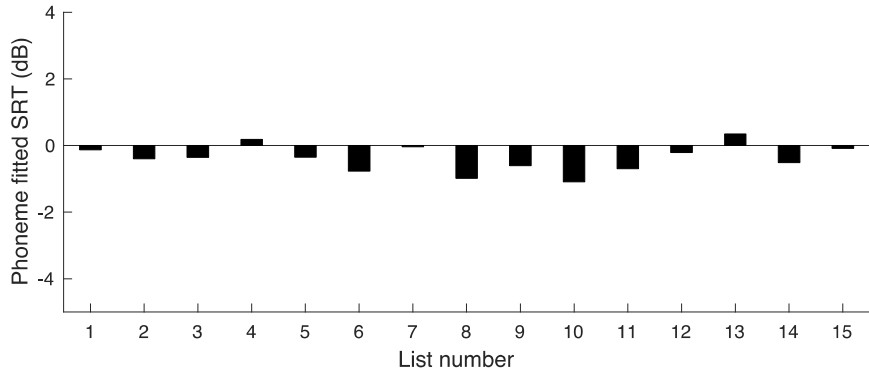

**Figure 4.** Results of experiment 2, showing SRT values corresponding to a phoneme score of 91.5 RAU for lists 2–16.

## 5. Experiment 3: Test–Retest Repeatability for Participants with Hearing Loss Tested in Quiet

In clinical practice, it might be desired to assess the ability of a person with hearing loss to understand speech in quiet at different sound levels. Experiment 3 assessed the test–retest repeatability of the COPT for participants with hearing loss tested without a background noise.

*5.1. Participants*

Fifteen participants with moderate-to-moderately severe bilateral sensorineural hearing loss were tested. Five were men and ten were women. Their ages ranged from 71 to 87 years with a mean of 80 years. Only the better-hearing ear was tested (based on the average threshold across 0.5 to 4 kHz). Eight were tested using the right ear and seven were tested using the left ear.

*5.2. Procedure*

The participant was seated in a soundproof room. Stimuli were played from a Cambridge Audio CD player via a Kamplex KC 35 clinical audiometer (Birmingham, UK) and presented to the better-hearing ear via TDH 39 headphones. No amplification was provided, to compensate for the hearing loss. List 1 was used for practice and to select an appropriate level for the main assessment of test–retest repeatability. The goal was to find a level that avoided floor and ceiling effects. The procedure for list 1 was as follows: A presentation level of 60 dB SPL was used for the first five words. If the participant got all five correct, the level was decreased by 10 dB. If the participant got three or four correct, the level was left unchanged. If the participant got zero, one, or two correct, the level was increased by 10 dB. The resulting level, denoted X, was used for the next five words. If the participant got all five correct, the level was decreased by 10 dB. If the participant got three or four correct, the level was left unchanged. If the participant got zero, one, or two correct, the level was increased by 10 dB. The resulting level, denoted Y, was used for the last five words. If the participant got all five correct, the main test was performed at Y − 10 dB. If the participant got three or four correct, the main test was performed at Y dB. If the participant got zero, one, or two correct, the main test was performed at Y + 10 dB.

Each participant was tested using two lists (test and retest) at the final selected level. The lists used with each participant were as follows:

Participant 1, lists 2 and 3;
Participant 2, lists 3 and 4;
Participant 3, lists 4 and 5;
.......
Participant 14, lists 15 and 16;
Participant 15, lists 16 and 2.

In this way, each list was used once for the test and once for the re-test.

*5.3. Results*

The mean test level was 49 dB SPL, with a range from 30 to 65 dB SPL and an SD of 9.9 dB. For words, the mean score in RAU was 59.2 (SD = 15.5) (corresponding to 60.5%) for the first test list and 58.0 (SD = 15.5) (59.1%) for the second test list. Based on a *t*-test, the difference was not significant ($p > 0.05$). Thus, there was no evidence for a practice or fatigue effect. To assess the test–retest reliability associated with the application of a single test list, critical differences were calculated, as for experiments 1 and 2. The critical difference for words was 19.0 RAU, corresponding to about three words.

For phonemes, the mean score in RAU was 78.1 (SD = 10.9) (79.0%) for the first test list and 80.1 (SD = 12.0) (80.8%) for the second test list. Based on a *t*-test, the difference was not significant ($p > 0.05$). Thus, there was no evidence for a practice or fatigue effect. The critical difference for phonemes was 13.7 RAU, corresponding to about six phonemes.

## 6. Discussion

The COPT open-set speech material was developed for use in clinical settings for patients familiar with British English using HAs or CIs. This study describes how the initial COPT materials were modified to reduce the variability across lists and provides information about practice effects and test–retest repeatability of the COPT for participants with normal hearing listening in the presence of speech-shaped noise and for participants with impaired hearing tested in quiet.

For the adjusted COPT materials used in experiment 2, the mean slope of the psychometric function relating RAU score to SNR was 4.4 RAU/dB. When expressed in percent, the slope varied with performance level. In the region of 80% correct, the slope was about 4%/dB. This is less than has typically been obtained for sentences presented in noise. For example, Plomp and Mimpen [19] reported a slope of about 15%/dB. The relatively shallow slope in percent/dB obtained here was partly a consequence of the fact that we used SNRs leading to relatively high performance levels, to use SNRs that are similar to those encountered in everyday life, whereas the procedure of Plomp and Mimpen [19], and many other procedures [20,32] track the SNR leading to 50% correct. Also, the slope may be greater for meaningful sentences than for single words [45].

Two methods of scoring for the COPT material were used, whole-word scoring and phoneme scoring. For the adjusted lists, the critical difference for speech in noise based on a single list per condition and normal-hearing participants was about 2 words (out of 15) or 4 phonemes (out of 50) for a significance level of $p < 0.05$. The critical difference for speech in quiet based on a single list per condition for participants with hearing loss was about 3 words (out of 15) or 6 phonemes (out of 50) for a significance level of $p < 0.05$. These critical differences are similar to or smaller than those found for other tests with a comparable number of items [10,12,46]. The time to administer a single list is about three minutes, which is sufficiently short for use in clinical practice. The combination of a short test time and small critical differences is an asset.

The finding that there were no practice effects after one practice list had been administered is important, since it means that improvements in COPT scores associated with the fitting or adjustment of a hearing device are likely to reflect true benefits of the device rather than practice effects. In clinical practice, when assessing the perception of speech in noise, there would be no need to estimate an SRT prior to testing at a fixed SNR. Instead, an appropriate SNR could be chosen based on results obtained while administering a practice list, similar to the method that was used for choosing an appropriate sound level in experiment 3, but perhaps with 3 dB steps in SNR rather than 10 dB steps in overall level.

The analysis of phoneme confusions made during open-set speech testing gives important information to the clinician about the individual client that may be useful for adjusting amplification characteristics. For example, if a target word of "cheese" is repeated as "shoes", this implies that the second formant of the vowel "ee" was not audible and the vowel was therefore heard as "oo". Also, the plosive burst of the affricate /ch/ was not audible, so that /ch/ was mis-identified as /sh/. These findings imply the need for greater audibility or discriminability of frequency components around and somewhat above 2000 Hz, which might be achieved by greater gain or by applying/adjusting frequency lowering in an HA [47,48] or by changing channel gains or the frequency-to-electrode mapping in a CI. Thus, phoneme-specific analysis might be useful for the fine-tuning of hearing devices as well as allowing the assessment of overall performance. However, further research is needed to assess the effectiveness of the COPT for the fine tuning of hearing devices.

In this study, the participants responded by writing down the word that they heard. This might be difficult for some clinical patients, for example, older people with limited dexterity or people who are not accustomed to using a keyboard. Potential solutions to this problem are as follows: (1) a clinician could write down the spoken response of the patient, although this has the risk of introducing biases; (2) an automatic speech recognition system could be used to identify the spoken response of the patient, as has already been done for some speech tests [49].

In all experiments, the stimuli were presented via headphones. In clinical use, it is envisaged that the COPT material would be presented in a sound field, so that patients could be tested using their HAs or CIs. If it were desired to test each ear separately, then a plug or earmuff could be used for the non-test ear and/or the hearing device for that ear could be turned off. Provided that the testing room has low reverberation, we expect the results to be similar for headphone and sound-field presentation. However, an

evaluation of the COPT based on patients using their HAs or CIs in a sound-treated room is clearly needed.

Some limitations of this study should be noted. Although the carrier phrase used with the COPT should allow the gain and noise reduction settings in HAs and CIs to approach their steady values, the duration of the carrier phrase may not have been sufficient to allow full "settling", given that the time constants of the processing sometimes exceed 1000 ms while the test word in the COPT started on average 780 ms after the start of the carrier phrase. Secondly, normative data derived using normal-hearing participants via speech-in-noise testing do not fully characterise the results that may occur for people with hearing loss, including CI users, when tested in the presence of noise. For example, the slope of the psychometric function relating percent correct to SNR may depend on the degree and pattern of hearing loss and on the type of hearing device that is used. Also, it may often be desired to assess clients with hearing loss without any background noise. The slope of the psychometric function relating performance to sound level in the absence of noise has not yet been assessed for the COPT. Despite these limitations, we feel that the COPT has the potential to be useful for assessing the benefits of hearing devices and for fine-tuning those devices.

## 7. Summary and Conclusions

This paper assesses practice effects, the variability across lists, and the test–retest repeatability of the COPT open-set speech test, which is intended for use in the clinical assessment of people with hearing loss and for the evaluation and fine-tuning of hearing devices. The final version has 15 COPT lists, each containing 15 target words preceded by the phrase "Say the word . . .". Following administration of a single practice list, no practice effects were evident.

Adjustments were applied to the initial COPT lists used in experiment 1 to reduce the variability across lists. Based on the adjusted materials used in experiment 2, for word scores, the slope of the psychometric function was 4.4 RAU/dB change in SNR. For phoneme scores, the slope was 3.4 RAU/dB. The critical difference between scores for two lists was two words or four phonemes. The SD of the SRTs (based on words) across lists was only 0.16 dB.

For the COPT materials presented in quiet, critical differences assessed for participants with hearing loss were about 3 words (out of 15) or 6 phonemes (out of 50) for a significance level of $p < 0.05$. We conclude that the COPT has good repeatability and practice effects are small.

**Author Contributions:** Design of the experiments, all authors; conduct of experiments, M.K. and J.M.; analysis, M.S.-C., M.K., J.M. and B.C.J.M.; adjustments to the COPT lists to balance difficulty across lists, J.M.; writing—original draft preparation, B.C.J.M.; writing—review and editing, M.K., J.M., M.S.-C. and T.R. All authors have read and agreed to the published version of the manuscript.

**Funding:** M.S.-C. was partly funded by the Medical Research Council (MRC) UK (Grant MR/S002537/1), and by the National Institute for Health and Care Research (NIHR, PGfAR 201608). M.K. was funded by the Royal British Legion Centre for Blast Injury Studies (RBL 03). The costs of the study were partially supported by Chear. The views expressed in this publication are those of the authors and not necessarily those of the NIHR or the MRC.

**Institutional Review Board Statement:** This study was approved by the Imperial College Research Ethics Committee (protocol code IC 4718 on 05/01/2021).

**Informed Consent Statement:** Informed consent was obtained from all subjects involved in the study.

**Data Availability Statement:** The raw data for this study are available from the corresponding author on reasonable request.

**Acknowledgments:** We thank Michael Stone for assistance in recording the COPT materials. We also thank three reviewers for helpful comments on an earlier version of this paper.

**Conflicts of Interest:** Author J.M. may sell the COPT test and associated software. The other authors report no conflicts of interest.

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
