# Peer review of "Development of New Open-Set Speech Material for Use in Clinical Audiology with Speakers of British English"

_audiolres, doi:10.3390/audiolres14020024_

Round 1

Reviewer 1 Report

Comments and Suggestions for Authors

This study on development and evaluation of speech test material is well designed and relevant for clinical audiologists. Although the limited tests with hearing impaired subjects (experiment 3, in quiet condition only) will have to be supplemented by more systematic evaluations with hearing aid and CI users in quiet and in noise, the successful procedure for reduction of variance in difficulty when using different lists provides a solid basis for further investigations.

I do have only minor comments related to the description of the methods. I did not understand why the recordings had to be downsampled to 16 kHz (L164) and whether the original recordings, sampled at 44.1 kHz would be available for future work.

From the description of experiment 1 and 2 it was not clear to me whether the sequence of words within lists was randomized. Since familiarization with the test material would normally prohibit repetition of word or sentence lists in open speech recognition tests, item randomization within lists might eventually alleviate this problem.

Word and phoneme scores were analyzed in this study, based on written responses of the test participants. Would this method be applicable in a clinical environment with elderly patients and which rules for typing error corrections or phoneme omissions would have to be specified for robust phoneme confusion analyses? These are potential questions for a follow-up study, eventually worth mentioning in the discussion section.

L12: the sixth institution (University of Erlangen-Nuremberg) is not connected to any of the authors (presumably Tobias Reichenbach)

Reviewer 2 Report

Comments and Suggestions for Authors

Reviewer 3 Report

Comments and Suggestions for Authors

This manuscript described the development and fine-tuning of an open-set speech test for use with speakers of British English. The authors indicated a clinical need for open-set speech material for clinical use with the British English population. The stated goal for this new clinical test was to evaluate speech recognition of hearing-impaired listeners using hearing aids or cochlear implants. The current experiment aimed to evaluate the test’s list equivalency and test-retest consistency in normal hearing listeners (tested in noise) and hearing-impaired listeners (tested in quiet without amplification). Results indicated some inequivalence of list difficulty in experiment 1, which was corrected in experiments 2 and 3. Test-retest consistency was high across all three experiments. In general, the manuscript is of high quality. The topic is appropriate for the journal, the methodology is rigorous, and the results are well analyzed. The manuscript may benefit from some minor adjustments in the introduction and discussion, so that this experiment is properly framed within the context of the larger project to develop this new clinical tool.

A considerable focus of the introduction and discussion sections is placed on the utility of the test for patients with hearing aids and cochlear implants, which is at odds with the methodology employed in the experiment. On lines 101-105, the authors explain the purpose of the paper and their intention to test normal-hearing adults. This is immediately followed by an explanation that they will be targeting SNRs based on the listening behaviors of hearing-impaired listeners (avoiding 50% correct), which has the benefit of working within the range where signal-processing algorithms are most effective (lines 106-109). These decisions and rationale make sense for clinical testing but are not well aligned with the current investigation of normal hearing listeners. Similarly, the decision to test hearing impaired listeners in quiet has clinical merit but does not address the proposed test’s potential utility with hearing aids or cochlear implants. This disconnect continues through much of the paper. On lines 143-144, the authors explain the purpose of the carrier phrase was to allow “any signal processing in hearing devices to settle,” but this is not a relevant detail for the methodology of this experiment, which does not involve hearing devices (see also lines 561-566). In the discussion (lines 514-520) the authors reiterate their goal of sound field testing with hearing devices and recognize that additional research is needed to support this use. This sentence overshadows the relevance of the current experiment in that process. Testing normal hearing adults is typical as a stage in development, and more of the discussion (and introduction for that matter) should be devoted to framing the current investigation of list equivalency and test-retest consistency, rather than touting the potential use of the test in the sound field with hearing devices. Testing with hearing devices can be highlighted as a future direction or long-term goal but plans for that work should be described as a next step now that list equivalency and test-retest consistency have been established.
